

# Bioinformatical analysis of eukaryotic shugoshins reveals meiosis-specific features of vertebrate shugoshins

Tatiana M. Grishaeva, Darya Kulichenko and Yuri F. Bogdanov

Laboratory of Cytogenetics, Department of Genomics and Human Genetics, N.I. Vavilov Institute of General Genetics, Moscow, Russia

## ABSTRACT

**Background:** Shugoshins (SGOs) are proteins that protect cohesins located at the centromeres of sister chromatids from their early cleavage during mitosis and meiosis in plants, fungi, and animals. Their function is to prevent premature sister-chromatid disjunction and segregation. The study focused on the structural differences among SGOs acting during mitosis and meiosis that cause differences in chromosome behavior in these two types of cell division in different organisms.

**Methods:** A bioinformatical analysis of protein domains, conserved amino acid motifs, and physicochemical properties of 32 proteins from 25 species of plants, fungi, and animals was performed.

**Results:** We identified a C-terminal amino acid motif that is highly evolutionarily conserved among the SGOs protecting centromere cohesion of sister chromatids in meiotic anaphase I, but not among mitotic SGOs. This meiotic motif is arginine-rich in vertebrates. SGOs differ in different eukaryotic kingdoms by the sets and locations of amino acid motifs and the number of α-helical regions in the protein molecule.

**Discussion:** These structural differences between meiotic and mitotic SGOs probably could be responsible for the prolonged SGOs resistance to degradation during meiotic metaphase I and anaphase I. We suggest that the "arginine comb" in C-end meiotic motifs is capable of interaction by hydrogen bonds with guanine bases in the minor groove of DNA helix, thus protecting SGOs from hydrolysis. Our findings support independent evolution of meiosis in different lineages of multicellular organisms.

## INTRODUCTION

After DNA replication in the S-phase of the cell cycle, the sister DNA molecules, chromatids, are held together until their disjunction occurs in anaphase of cell division. The phenomenon of holding chromatids together, called cohesion, depends on the complex of a four essential proteins named cohesins (SMC1, SMC3, Scc1/RAD21 and Scc3/STAG). They play a main role in cohesion, but the process depends on more than ten other proteins, including Wapl and Sororin (*Peters, Tedeschi & Schmitz, 2008*).

Corresponding author
Tatiana M. Grishaeva,
grishaeva@vigg.ru

Shugoshin (SGO) is one of them, and SGOs of different organisms have been state to make a family of more or less conserved proteins (*Watanabe, 2005*).

During cell division SGOs protect cohesion of centromere regions of sister chromatids up to the beginning of anaphase, while cohesion in chromosome arms is already lost as early as in prophase. This order of events is true for mitosis and the second division of meiosis (meiosis II), but not for the first meiotic division (meiosis I). Cohesion of sister centromeres is protected during metaphase and anaphase of meiosis I by some kind of meiosis-specific SGOs. As a result, sister chromatids are incapable to separate, and homologous chromosomes, each consisting of two sister chromatids, move in their stead to the cell poles in meiotic anaphase I. The chromosome number is thereby reduced to a haploid set. Thus, the usual SGO function is essential for somatic cell divisions through all ontogenesis, but not for meiosis I. Proper segregation of homologous chromosomes in meiosis I depends on expression of a meiosis-specific SGO (*Gutiérrez-Caballero, Cebollero & Pendás, 2012*). Specific SGO function is active during only one division cycle, while the somatic function is restored in meiosis II. What is the difference between somatic (mitotic) and specific meiotic SGO forms? Some structural differences have been reported for particular proteins in particular biological species, while general rules have not been found yet. We aimed on comparative analysis within a large pool of different SGOs, trying to find key structural differences between somatic and meiotic SGOs.

The protein that protects pericentric cohesion of sister chromatids in meiosis I has been discovered experimentally by two independent research groups (*Kitajima, Kawashima & Watanabe, 2004*; *Rabitsch et al., 2004*) when studying meiosis in the yeast *Schizosaccharomyces pombe* and termed SGO (Sgo1). A direct BLAST search for its orthologs in proteomes of other eukaryotes has revealed related proteins only in two fungi species, *Saccharomyces cerevisiae* and *Neurospora crassa.* In addition, Sgo2 has been identified as a Sgo1 paralog (a form that occurs in mitosis) in *S. pombe* (*Kitajima, Kawashima & Watanabe, 2004*; *Rabitsch et al., 2004*). A comparison of Sgo1 and Sgo2 has shown similarity for two protein regions, a conserved C-terminal basic region and a less conserved N-terminal coiled coil (*Kitajima, Kawashima & Watanabe, 2004*; *Rabitsch et al., 2004*). These two domains were identified earlier in MEI-S332 protein (SGO) of *Drosophila.* The C-terminal domain was shown to be crucial for centromere localization of MEI-S332 protein (*Kerrebrock et al., 1995*; *Tang et al., 1998*).

Another bioinformatical method has been employed to further search for orthologs, considering the domain protein structure. The search has revealed related proteins in conventional genetic model species: *Drosophila melanogaster*, the nematode *Caenorhabditis elegans*, the plant *Arabidopsis thaliana*, and mouse, as well as in humans (*Kitajima, Kawashima & Watanabe, 2004*; *Rabitsch et al., 2004*). Similar proteins have been found in 15 other eukaryotes, including fungi, animals, and plants (*Watanabe, 2005*; *Hamant et al., 2005*; *Gómez et al., 2007*; *Wang et al., 2011*; *Gutiérrez-Caballero, Cebollero & Pendás, 2012*; *Zamariola et al., 2014*). The proteins all have only short similar motifs at the C ends of their molecules, and their N-terminal regions show even lower conservation apart from a coiled-coil structure. There are six conserved amino acid

residues in the C-terminal region and only two in the N-terminal domain in these proteins. Therefore, this limited homology is functional (*Kitajima, Kawashima & Watanabe, 2004*).

Single SGO form, SGO1, occurs in some organisms (*S. cerevisiae, N. crassa,* and *Zea mays*), while two forms persist in some others, acting differently in mitosis and meiosis I. SGO1 plays a role of the meiotic form in the plant *A. thaliana* and fission yeast, while SGOL2 (SGO-like 2) plays the same role in vertebrates, including *Homo sapiens*. SGO1 and SGO2 differ in size and their role during mitosis and meiosis. It is interesting that in fission yeast and in vertebrates SGO2 is longer than SGO1, but in *Arabidopsis* SGO2 is shorter than SGO1. It means, the protein length does not correlate with the protein role in meiosis or mitosis.

Shugoshins work similarly in mitosis and meiosis, but interact with different partner proteins. In mitosis of higher eukaryotes, SGO1 is phosphorylated by kinase CDK1 (*Liu, Rankin & Yu, 2013*). Phosphorylated SGO1 acts as a homodimer to bind with one serine/threonine protein phosphatase PP2A-B' molecule and is then directed to pericentromeric heterochromatin (*Xu et al., 2009*; *Kateneva & Higgins, 2009*; *Tanno et al., 2010*). Its binding to chromatin requires kinase Bub1 and proteins of the **m**itotic **c**entromeric-**a**ssociated **k**inesin (MCAK) complex. Shugoshin-associated PP2A dephosphorylates one of the cohesin complex subunits, stromalin SA2/STAG, and thus protects cohesin from ESL1/Separase cleavage (*Sakuno & Watanabe, 2009*; *Yin et al., 2013*). Shugoshin-associated PP2A dephosphorylates Sororin, component of cohesin complex, as well. This process antagonizes Aurora B and Cdk1 thus protecting cohesion until metaphase (*Nishiyama et al., 2013*). SGO has additionally been identified as a conserved centromeric adaptor of the chromosomal passenger complex (CPC) (*Tsukahara, Tanno & Watanabe, 2010*). CPC is needed for proper chromosome segregation in mitosis (*Gutiérrez-Caballero, Cebollero & Pendás, 2012*). Another shugoshin-dependent mechanism for protecting cohesion has been proposed. According to the proposal, SGO antagonizes Wapl association with cohesin (*Hara et al., 2014*).

In meiosis, SGO2 is phosphorylated by kinase Aurora B and similarly binds as a homodimer with phosphatase PP2A and MCAK complex. The complex dephosphorylates kleisin REC8 which is another subunit of the cohesin complex, to protect it from separase (cysteine protease) (*Xu et al., 2009*; *Macy, Wang & Yu, 2009*; *Tanno et al., 2010*; *Clift & Marston, 2011*). The association with pericentromeric heterochromatin requires the specific HP1 protein (Swi6 in the yeast *S. pombe*) and histone H2A phosphorylation at one amino acid residue by kinase Bub1 (*Yamagishi et al., 2008*; *Kawashima et al., 2010*).

It was believed for a long time that SGOs are a conserved protein family whose members have approximately the same function, localization, and protein partners, but show a moderate similarity only within the N- and C-terminal regions when compared among different plants and animals (*Kitajima, Kawashima & Watanabe, 2004*; *Rabitsch et al., 2004*; *Watanabe, 2005*; *Hamant et al., 2005*; *Gómez et al., 2007*; *Wang et al., 2011*). However, recent studies have revealed substantial differences in both amino

acid sequence and certain accessory functions among SGOs from different species (*Gutiérrez-Caballero, Cebollero & Pendás, 2012*; *Zamariola et al., 2013*, *2014*).

Shugoshins are only conventionally classified as meiotic and mitotic. In yeast *S. pombe*, Sgo1 acts only in meiosis I indeed to protect centromeric cohesion, while Sgo2 occurs in both mitosis and meiosis, but performs other functions rather than protecting cohesion (*Rabitsch et al., 2004*; *Watanabe & Kitajima, 2005*; *Sakuno & Watanabe, 2009*). In *A. thaliana*, both of the SGO forms occur in meiosis, but only SGO1 protects cohesion. The mitotic function of SGO2 is still unclear (*Zamariola et al., 2013*; *Cromer et al., 2013*). In vertebrates, two SGOs forms occur in both mitosis and meiosis (*Sakuno & Watanabe, 2009*), SGOL2 protecting cohesion in meiosis I and playing many other roles (*Gregan, Spirek & Rumpf, 2008*; *Lee et al., 2008*; *Llano & Sherman, 2008*; *Sakuno & Watanabe, 2009*; *Clift & Marston, 2011*; *Gómez et al., 2014*). There is no consensus as to the mitotic function of SGOL2. The other form, SGOL1, is similarly found in all cells in mice (*Gregan, Spirek & Rumpf, 2008*). SGOL1 protects centromeric cohesion of chromatids in mitosis (*Watanabe, 2005*; *Kitajima et al., 2006*; *McGuinness et al., 2005*; *Gutiérrez-Caballero, Cebollero & Pendás, 2012*) and possibly has additional functions in many vertebrates (cited from *Gutiérrez-Caballero, Cebollero & Pendás, 2012*). In particular, mammalian SGOL1 is involved in maintaining centriole cohesion (*Macy, Wang & Yu, 2009*). However, one must note that there is another point of view. According to this point of view, centrosome defects observed in SGOL1-defective cells might be an indirect effect of a cohesion abnormality (*Dai, Kateneva & Higgins, 2009*).

In budding yeasts, the sole SGO Sgo1 plays a minor role in segregation of homologous chromosomes during meiosis I, but is important for the sister kinetochore bias toward a biorientation (*Kiburz, Amon & Marston, 2008*). According to *Kitajima, Kawashima & Watanabe (2004)*, Sgo1 plays an important role in mitosis as well. It is necessary for proper segregation of sister chromatids, but by another mechanism than protection of centromere cohesion in mitosis.

In *D. melanogaster*, MEI-S332 has been described as a meiotic SGO. Its role in mitosis is a matter of discussion. SGOs mutants in *D. melanogaster* do not show any mitotic defects, and this protein does not protect mitotic centromere cohesion (*Kerrebrock et al., 1995*). Therefore, MEI-S332 is not essential for mitosis. This does not exclude the possibility that it contributes to congression, kinetochore biorientation, or spindle assembly in a nonessential manner (*Nogueira et al., 2014*). SGO1 of *O. sativa* (*Wang et al., 2011*) and *Z. mays* (*Hamant et al., 2005*) are dispensable for mitosis.

Thus, meiotic and mitotic SGO forms are recognized only with respect to their main function of protecting centromeric cohesion and only in certain organisms. We still tried to identify the structural features that would allow pooling meiotic SGO in one functional group.

A problem of the origin and evolution of meiosis is discussed, including the variation and evolution of several specific meiotic proteins like recombination proteins, proteins of synaptonemal complex, etc. (*Marcon & Moens, 2005*; *Egel & Penny, 2007*; *Bogdanov, Grishaeva & Dadashev, 2007*; *Grishaeva & Bogdanov, 2014*). The objective of our work is accordingly to analyze the structural features of meiotic SGOs by a set of
bioinformatical methods, such as COBALT, CDART, MEME, COILS program, Mobile portal—charge, and Compute pI/Mw tool. In particular, we compared the extent of conservation among eukaryotic taxa for different SGO forms, classifying them by structure (SGO1 and SGO2) and by their function (meiotic and mitotic). Our long distance aim is to find any structural features of sugoshins, which permit them to function differently in meiosis and mitosis.

## MATERIALS AND METHODS

Shugoshin amino acid sequences were sought in the NCBI (http://www.ncbi.nlm.nih.gov/) and UniProtKB/TrEMBL (http://www.uniprot.org/uniprot/) databases. The search was performed by protein identifiers (IDs) reported for SGOs or by key words. Because data on several proteins (with different IDs) were available from experimental articles and the databases for each eukaryotic species, an essential step was comparing the proteins retrieved for each species and choosing one for further analysis. A multiple sequence alignment was made for each of the SGO forms from one organism with the COBALT program (Cobalt Constraint-based Multiple Protein Alignment Tool, http://www.ncbi.nlm.nih.gov/tools/cobalt/cobalt.cgi?CMD=Web).

The COBALT program was used as well for constructing phylogenetic trees. Parameters: maximal sequence difference 0.9, other parameters—by default. For constructing C-end tree we used as queries short tracks containing C-end motif (or C-end functional domain, if the motif could not be detected). In cases when the motif was located near the C-end of protein molecule, we used this C-end, adding fife amino acids (aa) upstream to determined motif, considering discrepancy of motif and domain coordinates. In cases, when some proteins have C-motif/domain displaced to the middle part of the molecule, we used protein fragments of no longer than 70 aa, starting from some aa located upstream of C-end motif/domain.

In total, 32 SGOs from the proteomes of 25 eukaryotic species were analyzed by bioinformatical methods. The list of species included three plants, 12 fungi, five invertebrates, and five vertebrates. Among more than 120 candidate proteins, we chose those that had been identified experimentally, recommended as SGOs of the given species, or were the closest to the full size protein (Table 1).

Conserved functional domains of SGOs were identified using the CDART program (Conserved Domain Architecture Retrieval Tool, http://www.ncbi.nlm.nih.gov/Structure/cdd/wrpsb.cgi?).

The set and order of conserved motifs in SGO molecules were determined using the MEME program (Multiple Em for Motif Elicitation, http://meme-suite.org/tools/meme) with the following parameters: maximal number of motifs, 100; motif distribution in sequences, any number of repetitions; motif width, 6–300 amino acid residues. Default values were used for other parameters. Figures summarizing the MEME results are schematic and only approximately show the actual motif sizes because of their great variation.

The secondary structure of the proteins under study (the probability that an α-helical structure is formed) was identified using the COILS program (Prediction of Coiled

**Table 1 Shugoshins chosen for further analysis, their names used in this work, sizes (amino acid residues, a.a.), and NCBI IDs.**

| Protein | NCBI ID | Size, a.a. | Protein | NCBI ID | Size, a.a. |
|---|---|---|---|---|---|
| **Fungi** | | | **Invertebrates** | | |
| Sgo1 Sp[a] | Q9P7A0.1 | 319 | SGO1 Dm (MEI-S332) | Q24141.1 | 401 |
| Sgo2 Sp | O13734 | 647 | SGO1 Ce | Q18412.2 | 307 |
| Sgo1 Sc | Q08490.1 | 590 | SGO1 Cb | Q60ZS1.1 | 306 |
| Sgo1 Nc | Q872U8.1 | 774 | SGO1 Bm | CDP98524.1 | 1107[b] |
| Sgo1 Ag | NP_984314.2 | 648 | SGO1 Ci | XP_002129751.1 | 426[b] |
| Sgo1 Mg | EAA54538.1 | 552[b] | **Vertebrates** | | |
| Sgo1 Mo | XP_003709333.1 | 544[b] | SGOL1 Xl (SGO-like) | NP_001090071.1 | 663 |
| Sgo1 Vv | KDB14582.1 | 621 | SGOL2 Xl | NP_001243696.1 | 1,029 |
| Sgo1 Tv | EHK16025.1 | 636[b] | SGOL1 Oh, partial | ETE65485.1 | 553 |
| Sgo1 Yl | CAG81849.1 | 823[b] | SGOL2 Oh, partial | ETE62590.1 | 874 |
| Sgo1 Tm | KFX44805.1 | 659 | SGOL1 Dr | NP_001074089.1 | 618 |
| Sgo1 An | Q5BDI1.1 | 479 | SGOL2 Dr | NP_001116771.1 | 847 |
| Sgo1 Cg | Q6FMT2.1 | 603 | SGOL1 Mm | Q9CXH7.1 | 517 |
| **Plants** | | | SGOL2 Mm | Q7TSY8.1 | 1,164 |
| SGO1 At | NP_187655.2 | 572 | SGOL1 Hs | Q5FBB7.1 | 561 |
| SGO2 At | NP_196052.2 | 419 | SGOL2 Hs | Q562F6.2 | 1,265 |
| SGO1 Os | ADO32586.1 | 486 | | | |
| SGO1 Zm | Q4QSC8.1 | 474 | | | |

**Notes:**
[a] Fungi: Sp, *Schizosaccharomyces pombe*; Sc, *Saccharomyces cerevisiae*; Nc, *Neurospora crassa*; Ag, *Ashbya gossipii*; Mg, *Magnaporthe grisea*; Mo, *Magnaporthe orizae*; Vv, *Villosiclava virens*; Tv, *Trichoderma virens*; Yl, *Yarrowia lipolytica*; Tm, *Talaromyces marneffei*; An, *Aspergillus nidulan*; and Cg, *Candida glabrata*; plants: At, *Arabidopsis thaliana*; Os, *Oryza sativa*; and Zm, *Zea mays*; an insect: Dm, *Drosophila melanogaster;* nematodes: Ce, *Caenorhabditis elegans*; Cb, *Caenorhabditis briggsae*; and Bm, *Brugia malayi*; an ascidian: Ci, *Ciona intestinalis*; vertebrates: Xl, *Xenopus laevis*; Oh, *Ophiophagus hannah*; Dr, *Danio rerio*; Mm, *Mus musculus*; and Hs, *Homo sapiens*.
[b] The protein has been annotated as predicted or hypothetical or otherwise. In all other cases, the protein is a conventional shugoshin.

Coil Regions in Proteins, http://www.ch.embnet.org/software/COILS_form.html) with a window width of 28 and default other parameters.

The static electrical charge distribution along a SGO molecule was studied using the "Mobile portal—charge" program of the Mobile Pasteur package (http://mobyle.pasteur.fr/cgi-bin/portal.py?#forms::charge) with the following parameters: window width, 25; data plotting, yes; and image format, png. Default values were used for other parameters.

The isoelectric point (pI) was determined using the Compute pI/Mw tool, which is available from the SIB Bioinformatics Resource Portal, ExPASy (http://web.expasy.org/compute_pi/).

## RESULTS

Choosing from NCBI and UniProtKB databases, we selected 32 proteins annotated as SGOs real or hypothetical, but obligatory with the term "SGO" in the annotation. Ten proteins were described in the literature as meiotic, seven proteins as mitotic, and

the remaining 15 proteins were non-specified. We have compared different groups of SGOs (meiotic vs mitotic, all SGO1, all SGO2 etc.) by a set of in silico methods.

We obtained a large massive of data. SGOs were analyzed according to several protein parameters mentioned in "Methods." We compared SGOs in different combinations (meiotic vs mitotic, only mitotic, only meiotic, only SGO1, only SGO2, both SGO1 and SGO2 and in other combinations). To describe the results of this analyzes, we choose a description principle as to compare all proteins in common by protein parameter analyzed, because some protein parameters were found as being similar in all the groups compared, while comparing of secondary structure and of some sequence motifs revealed some differences. It seems to us that the description of the results, as it follows below, i.e., according to the method employed, would be the most readable.

## Functional domains of shugoshins

The CDART program performs search of conserved domains in the database. "CDART finds protein similarities across significant evolutionary distances using sensitive domain profiles rather than direct sequence similarity" (cited from CDART annotation). CDART relies on annotated functional domains.

Knowing from the literature that the N- and C-terminal functional domains are moderately conserved among SGOs, we tried to apply an apparently formal procedure to identify the domains in the selected proteins (Table 1). However, CDART did not identify the domains in almost half of the proteins under study, including both proteins examined experimentally and predicted ones. The N-terminal functional domain was not detected in the SGO1 and SGO2 proteins of *A. thaliana* (Figs. 1a and 1d), as well as in SGO1 of *O. sativa*, and several invertebrate and vertebrate species (Table 2). The C-terminal domain was not found in the SGO1 proteins of *Drosophila*, certain fungi, and the snake *Ophiophagus hannah*. In the last case, the failure was likely explained by the fact that only a truncated protein variant was available from the databases for *O. hannah*. Both of the domains were not detected in human SGOL2 by CDART (Fig. 3d), although the protein has been annotated in the NCBI database as having an N-terminal coiled-coil SGO domain. The two domains were similarly not detected in chordate animal, ascidian *Ciona* SGO1. The ascidian protein has been predicted by bioinformatical methods, but has not been characterized in the NCBI annotation. In summary, meiotic and mitotic SGOs did not differ by this parameter. Hence, to clear up the situation we analyzed other parameters reflecting the properties of the proteins.

## Secondary structure of proteins

COILS program predicts secondary structure, i.e., the probability to form α-helical structure, within a given protein molecule. A study of the secondary structure of SGOs showed that the α-helix co-localizes with the N-terminal functional domain in the proteins that we found to have one (Table 2; Figs. 1 and 3). The only exception is *Z. mays* SGO1 where α-helix is located downstream the domain. We assumed on this ground that other α-helix-forming SGOs similarly possess an N-terminal functional domain. Thus, all of the proteins examined were assumed to be SGOs in fact and to possess an

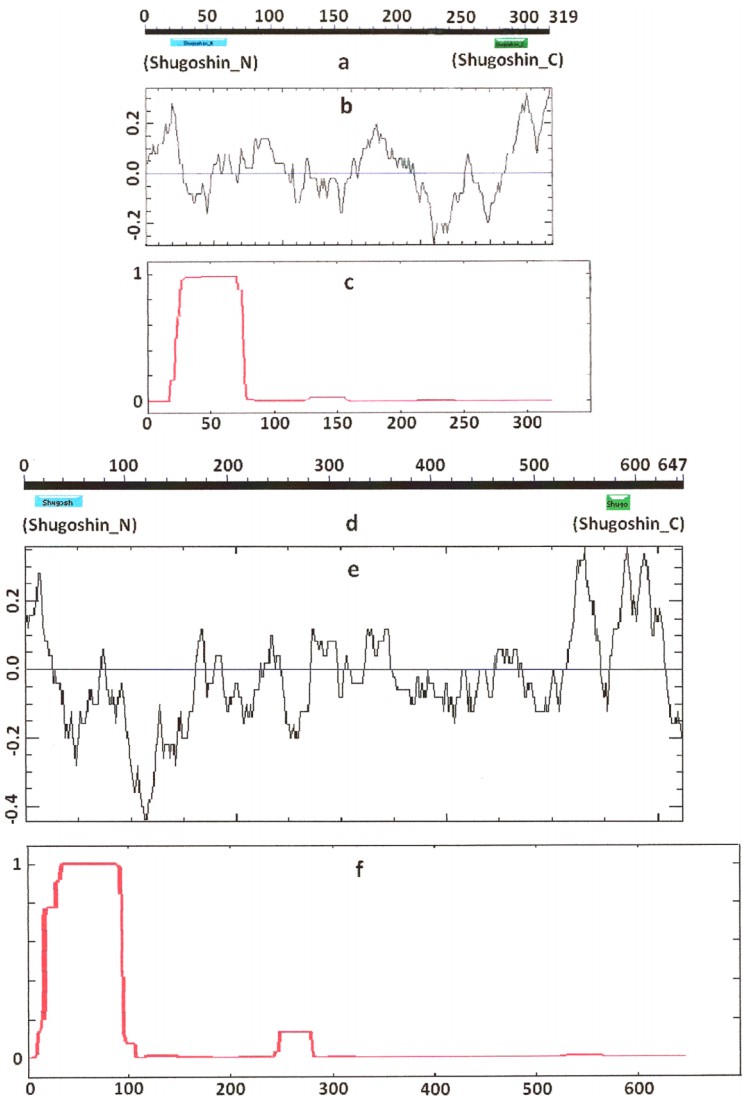

**Figure 1 Three parameters of *S. pombe* Sgo1 (meiotic) and Sgo2 (mitotic) shugoshins.** (a, d) Position of N- and C-terminal functional domains (results are obtained with CDART program), (b, e) electrostatic charge distribution along the protein molecules, and (c, f) probability for α-helical structure to be formed. Abscissa, amino acid sequence. Ordinate, charge (b, e) or probability (c, f). The relative lengths of proteins are not kept.

important N-terminal domain. The assumption pertains to meiotic and mitotic forms (Figs. 2 and 3), and all other SGOs.

Some SGOs have additional fragments of α-helix (Table 2). SGOL1 (vertebrates) have double N-end helices (Fig. 3c), and many SGOs have some fragments of helix at the central part of molecules. The last property is common almost for all SGO2 proteins but occurs in some SGO1s as well, independently of their function.

## Electrostatic charge distribution

Another important feature was revealed by studying the electrostatic charge distribution along protein molecule in the proteins under study. We determined an N-terminal positive

**Table 2 Comparison of shugoshins by four parameters (co-localization of the α-helix and the N-terminal domain, the presence of a positive charge peak ahead them, co-localization of the positive charge peak and the C-terminal domain, and the isoelectric point, pI).**

| Shugoshin, eukaryotic species, protein function | N-terminal region | | | C-terminal region | | | pI |
|---|---|---|---|---|---|---|---|
| | Functional domain | α-Helix | Positive charge peak | Functional domain | α-Helix | Positive charge peak | |
| Sgo1, Sp, meiotic | + | + | + | + | − | + | 8,91 |
| SGO1, At, meiotic | − | + | + | + | − | + | 9,25 |
| SGO1, Os, *meiotic* | −[a] | ± | + | + | ±[d] | + | 9,40 |
| SGO1, Zm, *meiotic* | + | +[c] | + | + | − | + | 9,39 |
| SGO1, Dm, *meiotic* | +[b] | + | − | − | − | | 8,80 |
| SGOL2, Xl, meiotic | − | + | + | + | +[d] | + | 7,35 |
| SGOL2, Oh, meiotic (partial) | − | +[b] | ±[e] | + | +[d] | + | 9,12 |
| SGOL2, Dr, meiotic | − | + | + | + | − | + | 6,55 |
| SGOL2, Mm, meiotic | + | + | + | + | ±[d] | + | 8,97 |
| SGOL2, Hs, meiotic | − | + | + | − | ±[d] | | 8,09 |
| Sgo2, Sp, mitotic | + | + | + | + | ±[d] | + | 5,15 |
| SGO2, At, mitotic | − | + | + | + | − | + | 9,56 |
| SGOL1, Xl, mitotic | + | ± (2) | + | + | ±[d] | + | 9,55 |
| SGOL1, Oh, mitotic (partial) | − | + (2)[b] | +[e] | − | − | | 6,26 |
| SGOL1, Dr, mitotic | − | + (2) | + | + | − | + | 9,87 |
| SGOL1, Mm, mitotic | + | + (2) | + | + | +[d] | + | 9,65 |
| SGOL1, Hs, mitotic | + | + (2) | + | + | +[d] | + | 9,27 |
| Sgo1, Sc, non-specified | + | + | + | +[f] | − | + | 9,26 |
| Sgo1, Nc, non-specified | + | + | + | +[f] | − | + | 8,09 |
| Sgo1, Ag, non-specified | + | + | + | +[fg] | − | + | 6,42 |
| Sgo1, Mg, non-specified | + | + | + | + | − | + | 9,93 |
| Sgo1, Mo, non-specified | + | + | + | + | − | + | 9,80 |
| Sgo1, Vv, non-specified | +[b] | + | − | −[g] | − | | 9,74 |
| Sgo1, Tv, non-specified | + | + | + | + | − | + | 9,40 |
| Sgo1, Yl, non-specified | + | + | ± | −[g] | +[d] | | 9,13 |
| Sgo1, Tm, non-specified | + | + | + | +[f] | − | + | 6,52 |
| Sgo1, An, non-specified | + | + | + | − | − | | 6,18 |
| Sgo1, Cg, non-specified | +[b] | + | − | +[f] | ±[d] | + | 9,22 |
| SGO1, Ce, non-specified | + | + | ± | + | − | + | 9,83 |
| SGO1, Cb, non-specified | + | + | ± | + | − | +[h] | 9,69 |
| SGO1, Bm, non-specified | −[a] | + | + | +[f] | − | + | 7,97 |
| SGO1, Ci, non-specified | − | + | + | − | − | | 9,17 |

**Notes:**
Shadowed are the most important protein characteristics. For details, see the text.
+, distinct peak; ±, small peak; (2), double peak.
[a] A domain other than the shugoshin domain co-localizes with the α-helix.
[b] The shugoshin domain or α-helix occur at the start of the molecule.
[c] An α-helix is located after the shugoshin domain.
[d] There are additional small α-helices in the central region of the molecule.
[e] A positive charge peak co-localizes with α-helix rather than precedes it.
[f] The shugoshin domain is displaced to the central region of the molecule.
[g] There is an additional domain(s) other than shugoshin domain.
[h] Peaks of positive charge are placed before and after the shugoshin domain.
The organisms are designated as in Table 1.

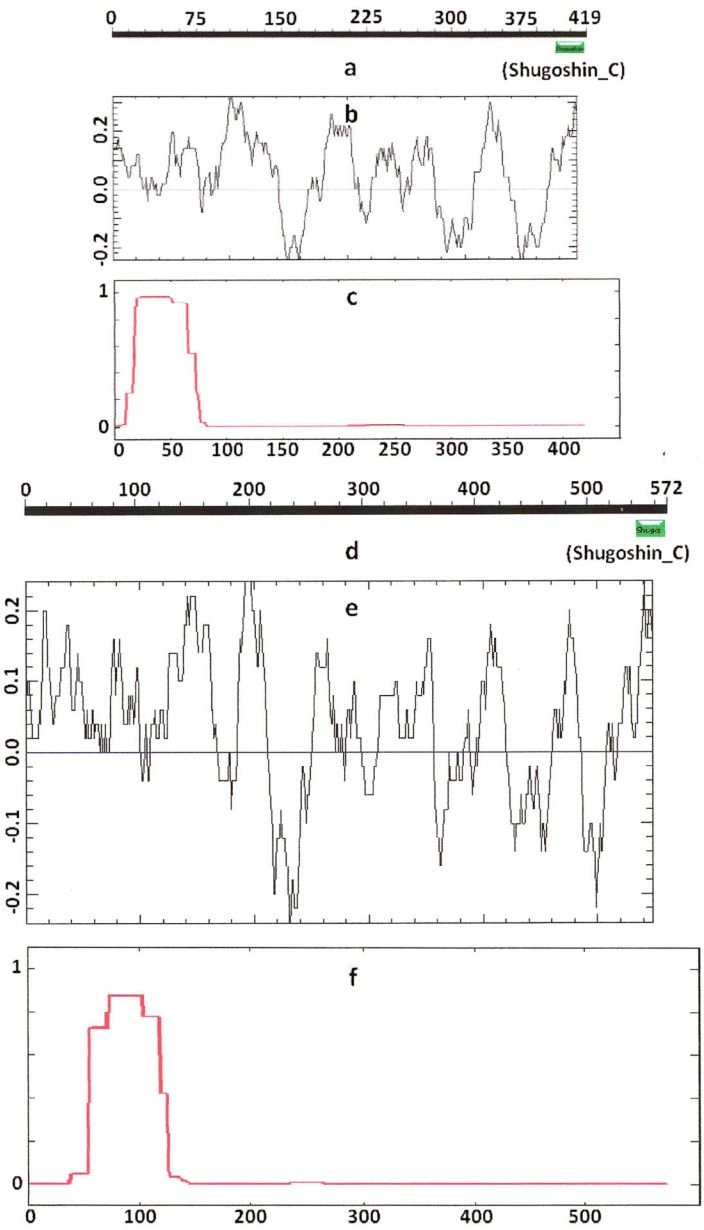

**Figure 2 Three parameters of *A. thaliana* SGO2 (mitotic) and SGO1 (meiotic) shugoshins.** Designations are as on Fig. 1. N-terminal functional domain is absent from both proteins (results are obtained with CDART program). The relative lengths of proteins are kept.

charge peak preceding the α-helical region in almost all of the proteins (Figs. 1–3; Table 2). Exceptions to this rule are observed when a protein molecule begins immediately with the domain and/or the α-helix. A positive charge peak either co-localized with the α-helix (in SGOL1 of *H. sapiens*, Fig. 3c; in both of the *O. hannah* SGOs) or was absent (in *Drosophila* and several fungal proteins) in this case. Meiotic SGOs did not differ in this feature from other SGOs. A positive charge peaks showed strong co-localization with the C-terminal domain in the proteins wherein the domain was identified by CDART (Figs. 1, 2 and 3b).

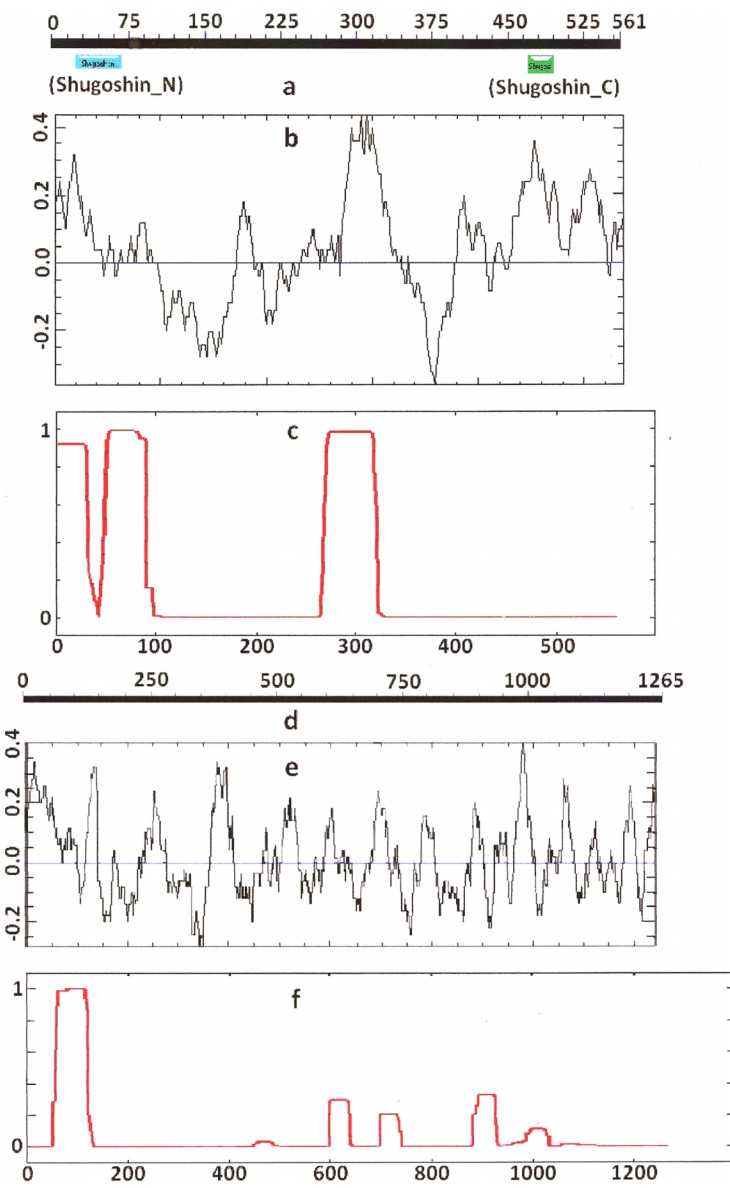

**Figure 3 Three parameters of *H. sapiens* SGOL1 (mitotic) and SGOL2 (meiotic) shugoshins.** Designations are as on Fig. 1. Both N- and C-terminal functional domains are absent in SGOL2 (results are obtained with CDART program). The formation of α-helices is observed not only in N-terminal, but also in the central region of both molecules. The relative lengths of proteins are not kept.

In the proteins wherein CDART failed to detect a C-terminal domain, the domain was impossible to predict by charge distribution because many positive charge peaks were observed along a SGOs molecule (Fig. 3e).

## Isoelectric point of proteins

The pI was another parameter used in the analysis. The parameter varied greatly, from 6.55 to 9.25, in meiotic SGOs (Table 2). Still, their pI values were within the variation range observed for other SGOs (from 5.15 to 9.87).

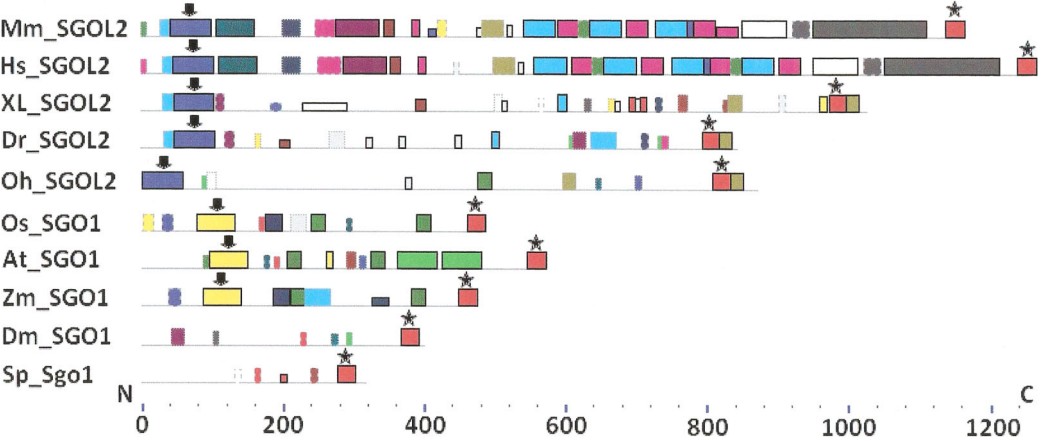

**Figure 4** Order of conserved amino acid motifs in meiotic shugoshins including those of *O. sativa*, *Z. mays*, and *D. melanogaster*. Species are indicated as in Table 1. A scale shows the amino acid sequence of a protein from the N toward the C end. Similar motifs are shown with bars of the same color and size. Results are obtained with MEME program. The only common motif is located at C end of proteins (asterisked). N-end motifs are different in vertebrates (blue bars with arrow) and plants (yellow bars with arrow). In fission yeast and in *Drosophila* no such motifs were found by MEME program.

## Conserved amino acid motifs in shugoshin molecules

CDART did not identify the domains in almost half of the proteins under study (see section "Functional domains of Shugoshins" of Results). In view of this situation, we decided to use other method to compare conservation of different groups of SGOs, namely, MEME program.

As it is stated by *Bailey et al. (2015)*, protein sequence motif is a short pattern that is conserved by evolution. Protein motif may correspond to the active site or some structural unit of a protein. Sequence motifs are basic functional units of molecular evolution. We analyzed conserved amino acid motifs in SGO molecules using MEME. Similar motifs *found* in proteins of different organisms are shown on Fig. 4 with bars of the same color and size. The same motifs on Fig. 5 have also similar colors, but different from Fig. 4, because Figs. 4 and 5 are drawn as results of separate queries to MEME program which makes colors automatically.

Only minor sequence similarity was observed for meiotic SGOs including (a) meiotic forms from the organisms that possess two SGOs forms, and (b) SGOs of *O. sativa*, *Z. mays*, and *D. melanogaster* that were shown to be truly meiotic. A C-terminal motif was the only motif traceable in all meiotic SGOs (Fig. 4, the motif is asterisked), and this is the first essential conclusion of our work. Common N-terminal motifs, which co-localize with functional domains, were found only in vertebrates (Fig. 4, the motifs are indicated with arrows). No other common motifs have been found except those in two mammals. Even a lower similarity in significant parts of molecules was found for mitotic SGOs from the organisms that possess two SGOs forms (Fig. 5). The common C-terminal motif was detected only in vertebrates excluding snake *O. hannah* (it has truncated protein). The N-terminal motif was observed also only in vertebrates, except *Danio rerio* (Fig. 5, arrows). Vertebrates possess some other common motifs (red and

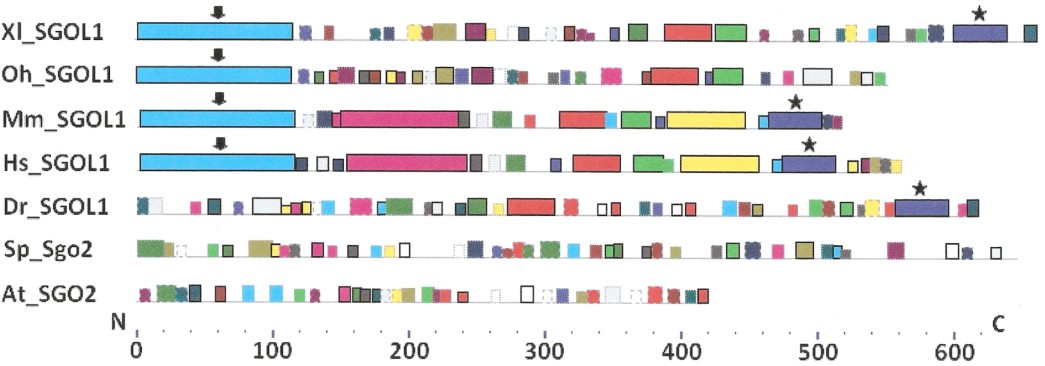

**Figure 5 Order of conserved amino acid motifs in mitotic shugoshins from species having two shugoshin forms.** Species are indicated as in Table 1. A scale shows the amino acid sequence of protein from the N toward the C end. Similar motifs are shown with bars of the same color and size. Results are obtained with MEME program. The common motif at C end (asterisked) is found only in vertebrates excluding snake *O. hannah* (it has truncated protein). N-end motif is found also in vertebrates (arrows) excluding fish *D. rerio*.

green bars on Fig. 5). They were found in the central part of molecules, which lack functional domains.

Localization of the C-terminal functional domain (CDART program) and the conserved amino acid sequence motif (MEME program, Fig. 4) were compared among meiotic SGOs. Nearly coincident coordinates were observed for the two structural elements. The maximal deviation was in one or two amino acid residues.

Then, we studied C-end motifs to investigate their conservation. The consensus sequence of the C-terminal motif for **meiotic** SGOs was identified by MEME program as **R**Y**RRRR**ACKPVSYKEPSL**R**CKM**RR**, being rich in arginine (**R**). We performed an analogous study with mitotic SGOs in seven species having two SGO forms (Fig. 5). The common C-terminal motif was detected only in vertebrates (SGOL1 in Mm, Hs, Xl, and Dr; asterisked), its consensus was identified for **mitotic** SGOs by MEME as K**RR**CTAAVNYKEPTLASKL**RR**GDPFTDLCFLNSPIFKQ, having less arginine residues.

For more detailed analysis we presented C-end motifs of meiotic and mitotic SGOs in Table 3. As soon as MEME program does not found C-end motifs in *S. pombe* and *A. thaliana* mitotic SGOs (Sgo2 and SGO2, respectively), we performed a new search. We used COBALT program and found the corresponding sequence in *S. pombe* as the result of alignment. At the same time, the method did not give results in *Arabidopsis*, may be, because of great size differences of mitotic and meiotic forms in this species. To continue the study in *Arabidopsis*, we compared C-end functional domains revealed by CDART program, taking into consideration that these structural units of SGO molecules almost co-localize. SGOs of plants other than *Arabidopsis*, as well as in *Drosophila*, do not have mitotic paralogs; hence, their pairwise comparison cannot be done. SGOL1 of *O. hannah* was found truncated and excluded from analysis.

Analyzing data presented in Table 3, we can make some conclusions. *S. pombe* and *Arabidopsis* have slightly more arginine residues at C ends of mitotic SGOs than in

**Table 3 Conserved C-end motifs (domains) in mitotic and meiotic shugoshins revealed by MEME (CDART) program.**

| Species, protein | Function[a] | Motif coordinates | Motif sequence |
|---|---|---|---|
| Sp Sgo1 | **Mei** | 278–301 | RE**KLRR**SVKVINYAIPSL**R**TKL**RR**[b] |
| Sp Sgo2 | Mit | 569–592[c] | DG**RSRRER**KKVNYALPGL**R**TKL**RR** |
| At SGO1 | **Mei** | 545–571[d] | VG**R**PS**R**QAAEKIKSYKEPSLKEKM**R**G |
| At SGO2 | Mit | 392–417[d] | VG**R**PS**R**HAAEKVQSY**R**EVSL**R**VKM**RR** |
| Os SGO1 | **Mei** | 462–485 | **R**PS**RR**AAEKIVSYKEVPLNIKM**RR** |
| Zm SGO1 | **Mei** | 450–473 | **R**SL**RR**AAEKVVSYKEMPLNVKM**RR** |
| Dm SGO1 | **Mei** | 368–391 | SA**R**PS**R**SC**R**PTSLVEPSLKNKL**R**N |
| Xl SGOL2 | **Mei** | 976–999 | ASS**RRKR**NPVNYKEPSLGTKL**RR**G |
| Xl SGOL1 | Mit | 602–639 | K**RR**CKVSINYAEPKLSGKL**RR**GDPFTDSEFLQSPIFKN |
| Oh SGOL2 | **Mei** | 810–833 | IYPS**RRR**KKDVSYAEPSLN**R**KL**RR** |
| Oh SGOL1 | Mit | Not found, the C-end truncated | |
| Dr SGOL2 | **Mei** | 796–819 | LG**R**P**RRR**ATPVTYKEPKINCKM**RR** |
| Dr SGOL1 | Mit | 558–595 | Q**RR**AASAVNYKEPSINTKL**RR**GDKFTDTRFLRSPIFKQ |
| Mm SGOL2 | **Mei** | 1,141–1,164 | PM**RRKR**QCVPLNLTEPSL**R**SKM**RR** |
| Mm SGOL1 | Mit | 465–502 | K**RR**CSTIKSYKEPTLASKL**RR**GDPFTDLCFLNSPIFKQ |
| Hs SGOL2 | **Mei** | 1,242–1,265 | **R**TS**RRR**CTPFYFKEPSL**R**DKM**RR** |
| Hs SGOL1 | Mit | 475–512 | K**RR**CTASVNYKEPTLASKL**RR**GDPFTDLCFLNSPIFKQ |

Notes:
[a] Protein function: mei, meiotic; mit, mitotic.
[b] Bolded are arginine residues.
[c] This stretch is found with the help of alignment by COBALT program because MEME did not found C-end motif (Fig. 5).
[d] C-end functional domains used (details see in the text).

meiotic ones. In common, their mitotic SGOs share a motif similar to their meiotic counterparts. Vice versa, the clearest enrichment in arginine residues is observed in meiotic SGOs of vertebrates. This is the second essential conclusion of our work.

## Analysis of all protein parameters

Two features were noted when comparing all SGOs forms (Figs. 4 and 5; Table 3). First, fungi and plants stood quite apart because even the C-terminal motifs of their proteins slightly differed from those in vertebrates. The N-terminal motifs were also different. This finding is supported by Cobalt tree (Fig. 6). This kind of tree is not phylogenetic tree, but reflects the results of multiple alignments of all 32 SGOs under our study. We can see distinct classes of SGOs belonging to fungi, nematodes, plants, and vertebrates. *Drosophila* MEI-S332 is clustered with fungal proteins. Clusters of meiotic or mitotic SGOs are absent but clusters of SGOL1 and SGOL2 are visible. We tried to obtain phylogenetic tree by methods of Fast Minimum Evolution or Neighbor Joining. The resulting tree was incomplete: Cobalt program have rejected about half of sequences being too different from the others, which have more similarity in their structure. Then we analyzed sequences neighboring C-end motifs/domains of 29 from 32 SGOs because three proteins have no detected C-end motifs/domains. The method used was Fast Minimum Evolution. SGOs of mouse were rejected by program, and one can see the

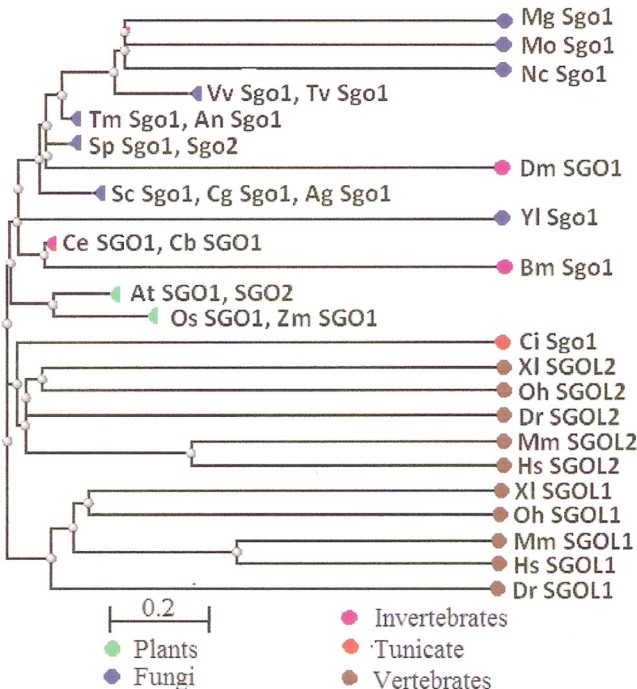

**Figure 6 COBALT tree for 32 shugoshin proteins.** The tree is constructed by COBALT program. Shugoshin names are indicated as in Tables. Almost all shugoshins are clustered according to their appurtenance to different multicellular lineages. The evolutionary distance between two sequences was modeled as expected fraction of amino acid substitutions per site given the fraction of mismatched amino acids in the aligned region (according to *Grishin, 1995*).

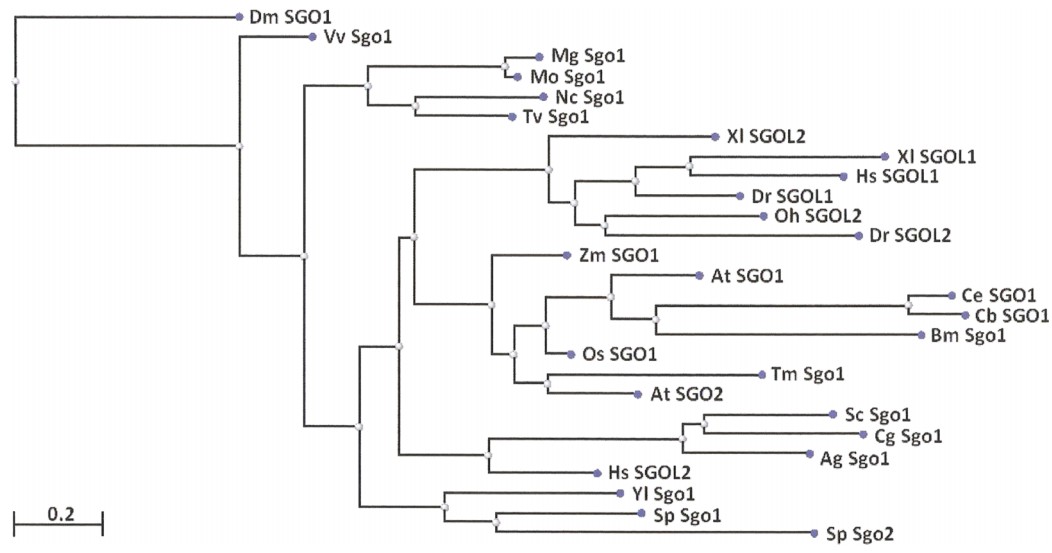

**Figure 7 Phylogenetic tree for 27 shugoshins.** The tree is constructed by COBALT program using the method "Fast Minimum evolution." Shugoshin names are indicated as in Tables. Two proteins (SGOL1 and SGOL2 of mouse) were automatically removed from the final version of the tree. Almost all shugoshins with their species are clustered according there appurtenance to different multicellular lineages. The evolutionary distance between two sequences was modeled as expected fraction of amino acid substitutions per site given the fraction of mismatched amino acids in the aligned region (according to *Grishin, 1995*).

resulting tree on Fig. 7. We see three clusters of fungi, cluster of vertebrates, and common cluster of plants, nematodes and one fungus. *Drosophila* protein is separated from others but is neighboring fungi as on the Fig. 6. Only SGOL2 of human is associated with some fungi. We cannot explain this casus. In common, two trees are almost in accordance with each other. Almost all SGOs and biological species are clustered according to their appurtenance to different phylogenetic lineages of Eukaryotes. This is the third essential conclusion of our work.

Taking into account other parameters of proteins (Table 2), one can note a far greater similarity within the SGO2 group and especially within the SGO1 group. As it is seen, pI values were high in the majority of the SGO1 proteins, amounting to nine or more, either to eight in fewer cases (Table 2, shadowed gray). The SGO2 proteins had lower pI values. On the other hand, additional structural fragments of α-helix, occurring in the central part of the molecule, were more common in the SGO2 protein group while double α-helices were found at the N end only in SGOL1 proteins (Table 2, shadowed gray). The N-terminal functional domain was more often undetectable by CDART in the SGO2 group, i.e., it was not strongly conserved. In summary, studied SGOs properties do not explain protein functions, keeping in mind their division into meiotic and mitotic ones, but rather correlate with their appurtenance to SGO1 or SGO2 group.

## DISCUSSION

In order to study the variation of SGOs in a broad evolutionary range of organisms, we employed a set of bioinformatics methods to analyze both structural and physicochemical features of the proteins. A diagnostic signature of all SGOs was identified; namely, a positively charged region which precedes an α-helix at the N end of the molecule. The signature was the most conserved among the SGOs from the 25 plant, fungal, and animal species examined in our work. We did not observe the signature only in the SGOs that had the α-helix at the very terminus of the molecule that was in an insect *D. melanogaster*, and fungi *Candida glabrata* and *Villosiclava virens* (Table 2).

Starting in silico study of SGOs, we observed that their N-terminal domains show an extremely low similarity even within a taxon. CDART failed to identify the N-terminal domain in 11 out of the 32 proteins in our set (Table 2). The observation is in line with the slight similarity reported for the N-terminal SGO domains in fungi in one of the earliest works on the SGO family (*Rabitsch et al., 2004*). The conventionally conserved SGO domain seems to vary greatly in primary structure among different eukaryotic kingdoms. Such a situation is not seldom with structural chromosomal meiotic proteins. For instance, proteins of the Scc1/RAD21/REC8 cohesin family differ in the set of conserved amino acid motifs even within the functional cohesin domain (*Bogdanov, Grishaeva & Dadashev, 2007*; *Grishaeva, Dadashev & Bogdanov, 2007*). The Scc3/SA/STAG stromalins, which belong to another cohesin family, are conserved only among vertebrates and show an extremely low similarity to their analogs found in early eukaryotes (*Grishaeva, Dadashev & Bogdanov, 2010*).

While the primary structure, i.e., amino acid sequence, is low conserved among SGOs, their secondary structure has features that are more typical. All members of the family

have a distinct α-helix at the N end (Figs. 1–3; Table 2). Two α-helical regions occur in tandem at the N end in the vertebrate SGOL2 proteins. In addition, α-helical regions are found in the central region of the SGOs molecule in vertebrates, the rice *O. sativa*, and certain fungi (Table 2). Thus, the secondary structure is conserved indeed in the SGO family, but the structural pattern is equally characteristic of both meiotic and mitotic SGOs.

We used different methods of protein analysis in our work. The results obtained complete one another. For example, the data in Figs. 3 and 4 present results obtained with different methods. At the first step of our work, we tried to reveal functional domains of SGOs with CDART program. However, this method turned to be not very sensitive and did not found domains in a number of SGOs analyzed (see Table 2). Figures 1–3 just illustrate nonconserved primary structure of SGOs but, at the same time, these figures illustrate much more conserved secondary structures.

Then, we used another instrument, the MEME program, to have possibility to compare all selected SGOs in species representing different phyla. MEME program reveals conserved motifs in molecules in query but these motifs may have no function or have yet unknown one. N- and C-end conserved motifs were revealed in all studied SGOs. Further, we were operating with these structures to compare degree of protein conservation. Figures 4 and 5 illustrate these conserved motifs.

The data in Fig. 5 and Table 3, concerning At SGO2 and Sp Sgo2, are results of searches done with different methods too. Figure 5 presents conserved motifs revealed with MEME program. However, C-end motifs in At SGO2 and Sp Sgo2 were not visualized by this method. This is why we used other methods to compare conservation of C-end parts of SGOs. Sp Sgo2 motif was found with COBALT alignment. In the case of Arabidopsis, we used not motifs but functional domains revealed with CDART program, taking into consideration that these structural units of SGOs molecules (i.e., functional domains and conserved motifs) almost co-localize.

Our analysis of the set of conserved amino acid motifs in SGOs, the charge distribution along the protein molecule, and pI values allowed us to conclude that the functional classification of SGOs into meiotic and mitotic lacks a structural basis apart from the fact that the meiotic proteins always have a small, highly conserved domain (or a motif when the domain is undetectable) at the C end (Fig. 4; Table 2). The C-terminal domain/motif is short, approximately 30 amino acid residues, but it is necessary for the exact SGO localization in the centromeric region of chromosomes (*Kitajima, Kawashima & Watanabe, 2004*; *Kawashima et al., 2010*). We analyzed many different SGOs (meiotic vs mitotic; only mitotic, either meiotic; or only SGO1, and only SGO2; or both SGO1 and SGO2 and many other combinations). However, 10 meiotic SGOs were found the sole group with common conserved C-end motif.

A greater similarity in several parameters is observed within other SGOs groups, SGO2 and especially SGO1. The most interesting features are shadowed gray in Table 2 and described in "Results." Fungi and plants stay apart because their SGOs display only a low primary structure similarity to vertebrate SGOs. SGOs apparently differ between different eukaryotic kingdoms.

As already mentioned in "Introduction," partitioning of SGOs into meiotic and mitotic groups is conventional, and SGOs are recognized as meiotic and mitotic only by their main function of protecting centromeric cohesion and only in some organisms. It seems that protecting cohesion is not the most important function in the case of SGOL2, and that other functions are of greater significance, being acquired during evolution by Sgo2 of primitive eukaryotes, such as *S. pombe*. New functions of this SGO developed with genome complication. For instance, human SGOL2 recruits kinesin MCAK to the centromere, where MCAK depolymerizes spindle microtubules attached in an improper manner. In *Xenopus laevis*, the same SGO regulates CPC-dependent spindle assembly (*Gutiérrez-Caballero, Cebollero & Pendás, 2012*).

In contrast, SGOL1 preserved the function of protecting cohesion, but only in mitosis. The function is of importance indeed, given that meiotic cohesion is dissolved via two steps in higher eukaryotes, first in chromosome arms (the so-called prophase pathway) and then in the centromere. Yet, the function was preserved by SGO1 and was not transferred to SGO2, as is evident from our findings. *Gutiérrez-Caballero, Cebollero & Pendás (2012)* have speculated that the original SGO function was protecting centromeric cohesion in meiosis and that the capability of protecting cohesion in mitosis was acquired by SGOL1 in vertebrates. As *Watanabe (2005)* notes, one could also speculate that the original function of SGO was to sense tension of microtubules by recruiting the CPC to centromeres.

## CONCLUSIONS AND PERSPECTIVES

Historically, SGOs were considered to be orthologs and to belong to a conserved family. However, recent studies showed that SGOs have a low amino acid sequence homology and display functional differences. Their functions should therefore be considered individually for yeasts, flies, and vertebrates. In spite of their common name, SGOs lack direct orthology and are highly diverse in amino acid sequence and functions (*Gutiérrez-Caballero, Cebollero & Pendás, 2012*).

Thus, any information obtained by comparing the SGO structure for different organisms is of importance for understanding the actual functions and mechanisms of action of SGOs. The conserved domain/motif in the C-terminal region of SGOs (*Kitajima, Kawashima & Watanabe, 2004*; *Kawashima et al., 2010*) turned out to be not so conserved as it was expected. Our results are of particular interest in this respect, demonstrating that conservation exist mostly among meiotic SGOs, but not among mitotic ones. This structural difference in meiotic and mitotic SGOs, probably, can be responsible for resistance of SGO against degradation during meiotic metaphase I and anaphase I, providing differences in sister-chromatids behavior in meiosis I and mitosis.

One can make some essential conclusions of our work.

1. Meiotic SGOs have common C-end motif conserved in fungi, plants and animals.

2. Mitotic SGOs possess less conserved C-end motif.

3. C-end motifs in meiotic SGOs of vertebrates are enriched in arginine residues.

4. Almost all SGOs and biological species are clustered according to their appurtenance to different phylogenetic lineages of Eukaryotes.

Two directions of further investigation could be proposed. One is to test capability of meiotic SGOs to interact with other accessory proteins that could protect SGOs from degradation during meiosis I. Another way is to pay attention to possible association of meiosis-specific arginine-rich motif of SGOs (found in our study) with centromere DNA during meiosis I. *Seeman, Rosenberg & Rich (1976)* hypothesized that, in the DNA minor groove, an arginine side group can form hydrogen bonds with a guanine base. Thus, several clustered arginine residues in a meiosis-specific motif of SGO can make "arginine comb" interacting with guanine bases in centromere DNA. Indeed, there is qualitative observation that arginine readily has high affinity to DNA. Recent thorough analysis (*Suvorova, Korostelev & Gelfand, 2015*) confirms this conclusion. In this case, the arginine-guanine association could be involved in protection of meiotic SGOs from degradation in meiosis I at least in vertebrates, while SGOs of mitotic chromosomes, which have a motif with less number of arginine residues, lakes such kind of association.

We conclude that meiotic SGOs are combined in one family by their function rather than by parameters characterizing their structure. Our results additionally indicate that either SGO1 or SGO2 evolved to act as a main meiotic form, the choice being made independently in different multicellular lineages, designated by *Cock, Akira & Susana (2011)*, namely, red and brown algae, green algae/plants, fungi, and animals, and being determined by a yet unclear factor (possibly, the capability of interacting with other accessory proteins).

## ACKNOWLEDGEMENTS

We thank Drs. V.G. Tumanyan and N.G. Esipova, Engelhardt Institute of Molecular Biology, Moscow, for valuable information about arginine-guanine interaction and appropriate references.

### Funding

The work was supported by the Russian Foundation for Basic Research (project no. 16-04-01447-a). The funders had no role in study design, data collection and analysis, decision to publish, or preparation of the manuscript.

### Grant Disclosures

The following grant information was disclosed by the authors:
Russian Foundation for Basic Research: 16-04-01447-a.

### Competing Interests

The authors declare that they have no competing interests.

### Author Contributions

- Tatiana M. Grishaeva performed the experiments, analyzed the data, contributed reagents/materials/analysis tools, wrote the paper, prepared figures and/or tables, reviewed drafts of the paper.

- Darya Kulichenko performed the experiments, contributed reagents/materials/analysis tools.
- Yuri F. Bogdanov conceived and designed the experiments, wrote the paper, reviewed drafts of the paper.

### Data Deposition

The raw data is included in Tables 2 and 3.

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
