# Peer review of "Bioinformatical analysis of eukaryotic shugoshins reveals meiosis-specific features of vertebrate shugoshins"

_PeerJ, doi:10.7717/peerj.2736_

## Round 0.1 · original submission · Major Revisions

Two experts in this field raised a number of problems on the manuscript to be solved before publication. I hope you find the comments valuable and revise the manuscript according to their comments.

Reviewer 1 ·

Basic reporting

Figures are not sufficient to understand the content of the article.

Experimental design

See the general comments.

Validity of the findings

See the general comments.

Additional comments

This manuscript describes bioinformatics comparison of shugoshin proteins from yeasts, plants and animals. The authors compared several parameters of the shugoshin proteins in relation with their functions or origins of species. The major difference between meiotic and mitotic shugoshins was attributed to the C-terminal amino acid residues. After all, it wasn’t clear whether the physicochemical properties are correlated to their functions. It is difficult to read the manuscript mainly because results are not organized systematically in correlation with functional classification, species or physicochemical properties. The manuscript contains potentially interesting materials, but needs to be revised to improve the readability.

1. In Table 1, provide information about classification of species (fungi, plants, invertebrates and vertebrates). Although species are described in the legend, it will be more helpful if the table is divided to fungi, plants, invertebrates and vertebrates.

2. In Table2, shogoshin proteins should be listed systematically according to their functions (meiotic, mitotic and non-specified). In this table, functional classification seems to be important to demonstrate correlation between functions and the parameters of interest (alpha-helix, charge distribution, pI, etc). Thus, it is recommended that shogoshin proteins are first categorized by their functions (meiotic, mitotic and non-specified), and then in each category, listed in the order of appearance in Table 1.

3. The authors identified the C-terminal motif in meiotic shugoshins: RYRRRRACKPVSYKEPSLRCKMRR. Provide alignment of amino acid sequences of meiotic shugoshins compared to deduce the consensus sequence.
In order to claim that this sequence is specific to meiotic shugoshins, comparison between meiotic and mitotic shugoshins in the same species is necessary.
The same thing for the consensus sequence specific to vertebrates. Provide alignment of amino acid sequences of meiotic shugoshins compared to deduce the consensus sequence: KRRCTAAVNYKEPTLASKLRRGDPFTDLCFLNSPIFKQ.

4. With respect to comparison between mitotic and meiotic shugoshins, it is recommended to show the properties of both meiotic and mitotic forms of S. pombe shugoshin in Figure 1. Similarly, compare meiotic and mitotic forms of A. thaliana in Figure 2, and meiotic and mitotic forms of H. sapiens shugoshin in Figure 3.

5. The authors need more efforts to explain Figure 4 and 5 for comparison of meiotic and mitotic shugoshins. If possible, a phylogenic tree of 32 shugoshin proteins (either full-lengths or N- and/or C-terminal portions) will be helpful.

Minor comments:
Page 2, line16: Change “the small groove” to “a minor groove”.
Page 2, line 41: Change “as the result” to “as a result”.
Page 3, line 29: “yeast” S. pombe might be better changed to “fission yeast” to distinguish it from budding yeasts because the authors call other yeasts as “budding yeasts” on page 4, line 25.

Reviewer 2 ·

Basic reporting

The authors performed a bioinformatic analysis of the shugoshin protein family. The analysis reveals some diversity of shugoshin family proteins, which is indeed informative for researchers in this field. The following points should be properly addressed before publication.


P3
L.35, ...less conserved protein. This should cite the original review about the discovery of shugoshin (Watanabe Curr Opin Cell Biol. 17, 590-595(2005)

L. 37-40, Note that shugoshin-associated PP2A dephosphorylates sorolin as well (Nishiyama PNAS 110, 13404, 2013, Lui NCB 15, 40-49, 2013). Moreover, another shugoshin-dependent cohesion protection mechanism has also been proposed: Shugoshin antagonizes Wapl association with cohesin (Hara NSMB, 21, 864-870, 2014).

L. 32, 33 Shugoshin is phosphorylated by CDK1 and binds cohesin (Lui NCB 15, 40-49, 2013).

L.48, The association with pericentromeric heterochromatin requires the specific HP1 protein (Swi6 in the yeast S. pombe) and histone H2A phosphorylation at one amino acid residue by kinase Bub1 (Sakuno, Watanabe, 2009; Macy et al., 2009).
This sentence should cite the key original papers [Yamagishi Nature 455, 251-255, (2008), Kawashima 327, 172-177 Science (2010)].

P4
L. 23, In particular, mammalian SGOL1 is involved in maintaining centriole cohesion. This is a very controversial issue. There are several papers that argue against a shugoshin function at centrosomes. Centrosome defects observed in SGOL1-defective cells would be an indirect effect of a cohesion defect (e.g. Dai et al. JCS122, 4168-4176, 2009; Leber et al. Sci Trans Med 2, 33ra38, 2010).

L. 27, Fission yeast Sgo1 is expressed only in meiosis and is genetically dispensable for mitosis (Kitajima et al 2004).

P7
L. 45-47, The C-terminal …(Tang 1998). This cites a wrong reference. It should be (Kitajima, et al. Nature 2004; Kawashima, et al. Science 2010).

P8
L. 18-19, It should be noted that the original shugoshin function might be sensing tension by recruiting the CPC to centromeres, and this function might extend to protect cohesin (see conclusion in Watanabe COCB 2005).

Experimental design

The comparison between mitotic and meiotic shugoshin is not properly performed. More precise comparison is required. The description in p6 L41-49 is not acceptable at all.

Validity of the findings

The conclusion might be not validated as mentioned in the 'General comments for the Authors".

Additional comments

The current manuscript ignores the most conserved motif of shugoshin, the C-terminus SGO motif defined in (Kitajima et al Nature 2004; Kawashima et al Science 2010). This should be clearly described. This motif is conserved between mitotic and meiotic (or all) shugoshin proteins.

---

## Round 0.2 · Major Revisions

Both of the reviewers appreciate that your manuscript has been greatly improved. However, they still find some problems to be solved. I agree with their opinions and ask you to revise the manuscript according to their suggestions. I hope their comments/suggestions are helpful to improve the paper.

Reviewer 1 ·

Basic reporting

See the general comments.

Experimental design

No concerns.

Validity of the findings

See the general comments.

Additional comments

Revisions have improved readability of the manuscript by organizing the results in a more systematic manner. This reviewer has one concern about their conclusions about conserved amino acid motifs in shugoshin molecules (page7 line25 – page 8 line 14). It doesn’t seem that these descriptions can be concluded by the contents of Table 3. The most straightforward conclusions of Table 3 may be:
(1) Meiotic shugoshins have a motif conserved in fungi, plants and animals
(2) Mitotic shugoshins in vertebrates have a less conserved motif.
(3) Mitotic shugoshins in fungus (Sp) and a plant (At) share a motif similar to their meiotic counterparts.
(4) As mitotic shugoshins in invertebrates are not presented, no comparisons can be done.
This concern is also related to the title. The title doesn’t make sense. If the above conclusions are correct, a possible title would be “Mitotic shugoshins in vertebrates are characterized by an arginine-rich motif less conserved than the conserved motif in meiotic shugoshins”. However, this section is difficult to follow, the authors should describe the results shown in Table3 more clearly before they can finalize the title.

Minor issues:
1. It seems that the authors mean “arginine-rich” by “arginine-reach” in the title, page2 line11 and page10 line35.

2. There are some typographic or grammatical errors, e.g., “in meiotic” probably missing an objective noun (page 8 line 13).

Reviewer 2 ·

Basic reporting

The manuscript was improved especially in Table 3. However, I still have some concerns as following.

I do not understand the meaning of the new title.

The C-end motif is absent in Sp-Sgo2 amd At-SGO2 in Figure 5. This result is not consistrent with the data in Table 3 as well as the previous two papers. Thus, this is very misleading.

In Figure 3, Shugoshin-N and Shugoshin-C are absent in SGOL2. Is this true? They are present in Figure 4.

Experimental design

No comments

Validity of the findings

Although the authors claim that the C-end motif is more conserved in meiotic shugoshin (Fig 4, 5), this is not consistent with the previous reports. In fact, the conserved C-end motif (Table 3) has been already clearly defined; the C-end motif is similarly conserved between Sgo1 and Sgo2 (or meitoc and mitotic shugoshins) in single organism (Kitajima et al. Nature 2004; Rabitsch et al. Cur Biol 2004). In Table 3 of this manuscript, the C-end motif of meiotic shugoshin possesses more arginines only in vertebrate but not in Arabidopsis and fission yeast. Thus, it is hard to claim that ‘(Results) we identified a C-terminal arginine-reach amino acid motif that is highly evolutionarily conserved among the shugoshins protecting centromere cohesion of sister chromatids in meiotic anaphase I, but not among mitotic shugoshins’.

Additional comments

No comments

---

## Round 0.3 · accepted · Accept

I am happy to inform you that your paper is accepted - both reviewers are satisfied by your revision.

Reviewer 1 ·

Basic reporting

The problems I pointed out have been solved.

Experimental design

No concerns.

Validity of the findings

The problems I pointed out have been solved.

Additional comments

Revisions have improved the manuscript.

Reviewer 2 ·

Basic reporting

No commnets

Experimental design

No commnets

Validity of the findings

No commnets

Additional comments

Now, the authors can publish the unique view about shugoshin family protein.